# Biocompatible Polymer-Grafted TiO_2_ Nanoparticle Sonosensitizers Prepared Using Phosphonic Acid-Functionalized RAFT Agent

**DOI:** 10.3390/polym15112426

**Published:** 2023-05-23

**Authors:** Yukiya Kitayama, Aoi Katayama, Zhicheng Shao, Atsushi Harada

**Affiliations:** 1Department of Applied Chemistry, Graduate School of Engineering, Osaka Prefecture University, 1-1 Gakuen-cho, Naka-ku, Sakai 599-8531, Osaka, Japan; 2Department of Applied Chemistry, Graduate School of Engineering, Osaka Metropolitan University, 1-1 Gakuen-cho, Naka-ku, Sakai 599-8531, Osaka, Japan

**Keywords:** sonodynamic therapy, TiO_2_, reversible addition–fragmentation chain transfer polymerization, phosphonic acid

## Abstract

Sonodynamic therapy is widely used in clinical studies including cancer therapy. The development of sonosensitizers is important for enhancing the generation of reactive oxygen species (ROS) under sonication. Herein, we have developed poly(2-methacryloyloxyethyl phosphorylcholine) (PMPC)-modified TiO_2_ nanoparticles as new biocompatible sonosensitizers with high colloidal stability under physiological conditions. To fabricate biocompatible sonosensitizers, a grafting-to approach was adopted with phosphonic-acid-functionalized PMPC, which was prepared by reversible addition–fragmentation chain transfer (RAFT) polymerization of 2-methacryloyloxyethyl phosphorylcholine (MPC) using a newly designed water-soluble RAFT agent possessing a phosphonic acid group. The phosphonic acid group can conjugate with the OH groups on the TiO_2_ nanoparticles. We have clarified that the phosphonic acid end group is more crucial for creating colloidally stable PMPC-modified TiO_2_ nanoparticles under physiological conditions than carboxylic-acid-functionalized PMPC-modified ones. Furthermore, the enhanced generation of singlet oxygen (^1^O_2_), an ROS, in the presence of PMPC-modified TiO_2_ nanoparticles was confirmed using a ^1^O_2_-reactive fluorescent probe. We believe that the PMPC-modified TiO_2_ nanoparticles prepared herein have potential utility as novel biocompatible sonosensitizers for cancer therapy.

## 1. Introduction

Ultrasounds with wavelengths beyond human hearing have been widely used in diagnosis and therapy because they can penetrate deep into tissues without radiation damage. Sonodynamic therapy (SDT) has been widely used in clinical studies, including cancer therapy, owing to its non-invasiveness and temporal-spatial controllability with great depth [1,2,3,4,5,6,7]. In SDT, reactive oxygen species (ROS) such as singlet oxygen (^1^O_2_) and hydroxyl radicals are generated under ultrasound irradiation, and the ROS induce oxidative damage to target tissues. Furthermore, the generated ROS breaks the redox balance in living cells, which induces effective treatment on the hypoxic tumor [8]. The mechanism of ROS generation is the cavitation effect induced by ultrasounds which causes sonoluminescence, and the phenomenon is attributed to generate ROS from sonosensitizer. For efficient SDTs, sonosensitizers are of great importance to initiate a sonochemical reaction when producing ROS. In the past, many sonosensitizers, such as TiO_2_ [9,10], porphyrin and its derivatives [11,12,13,14], BaTiO_3_ [15], and PtCu_3_ [16,17] have been developed to enhance ROS generation and therapeutic effects. TiO_2_ nanoparticles have the potential for targeted delivery to tumors through enhanced permeability and retentivity effects [18,19] owing to their nanometer size [5]. TiO_2_ nanoparticles have high chemical and physical stabilities; however, the colloidal stability of TiO_2_ nanoparticles under physiological conditions is poor. TiO_2_ nanoparticles have an isoelectric point at pH 6.2 when TiO_2_ is formed as an anatase crystal [20]. Thus, TiO_2_ nanoparticles have an anionic surface charge at neutral pH and are colloidally stabilized via electrostatic repulsion owing to their anionic charge. However, the electric bilayer on the particles becomes thinner in media containing high ionic concentrations, resulting in poor colloidal stability of the TiO_2_ nanoparticles. An important requirement to ensure the efficacy of the sonodynamic therapy is the high colloidal stability of TiO_2_ nanoparticles under physiological conditions.

Several approaches have been reported to prepare TiO_2_ nanoparticles with high colloidal stability for SDT under physiological conditions. Poly(ethylene glycol) (PEG) modification of TiO_2_ nanoparticles is used to create highly stable TiO_2_ nanoparticles for various therapies because PEG induces the steric repulsion of the PEG-modified nanoparticles [5,21]. Polysaccharide (e.g., dextran and hyaluronic acid)-modified TiO_2_ nanoparticles have also been prepared for sonodynamic cancer therapy [10,22]. In our previous studies, polyion complex (PIC) micelles incorporating TiO_2_ nanoparticles were developed as novel sonosensitizers possessing high colloidal stability under physiological conditions [23]. The PIC micelles were prepared from cationic polyallylamine-grafted poly(ethylene glycol) (PAA-*g*-PEG) and anionic TiO_2_ nanoparticles at neutral pH, where the micellar structure was stabilized via electrostatic and van der Waals interactions between the polyallylamine and TiO_2_ nanoparticles. The PIC micelles possessed high colloidal stability owing to the steric repulsion derived from the PEG grafted on the PIC micelles. Furthermore, we confirmed the sonosensitizing effect of PIC micelles incorporating TiO_2_ nanoparticles in vitro with HeLa cells, where decreased cell viability was observed in cells treated with PIC micelles by ultrasound irradiation compared to that in the untreated cells [24].

Recently, reversible deactivation radical polymerization (RDRP) has been developed as an emerging synthetic method to achieve precise molecular design and create various functional polymers via radical reaction [25,26,27,28,29,30,31,32,33,34,35,36,37,38,39,40,41,42,43,44]. Using RDRPs, the surface modification of inorganic nanoparticles can be easily achieved using *grafting-to* or *grafting-from* approaches [45,46,47,48,49,50]. Charpentier et al. reported the surface-initiated reversible addition–fragmentation chain transfer (RAFT) polymerization of methyl methacrylate from TiO_2_ nanoparticles modified with 4-cyano-4-(dodecyl-sulfanylthiocarbonyl) sulfanyl pentanoic acid [51]. Wang et al. reported the successful surface-initiated atom transfer radical polymerization (ATRP) of styrene from TiO_2_ nanoparticles using an initiator possessing a trimethoxysilane group [52]. Charpentier et al. reported water-dispersible poly(acrylic acid)-modified TiO_2_ nanoparticles by RAFT polymerization from TiO_2_ nanoparticles modified with 4-cyano-4-(dodecyl-sulfanylthiocarbonyl)sulfanyl pentanoic acid [53]. Haddleton et al. reported surface modification of 2-(dimethylamino)ethyl methacrylate and 2-(diethylamino)ethyl methacrylate on TiO_2_ nanoparticles using *grafting-to* or *grafting-from* approaches with Cu(0)-mediated living radical polymerization using a designed initiator containing catechol as a binding site to TiO_2_ [54]. However, to the best of our knowledge, the preparation of biocompatible polymer-modified TiO_2_ nanoparticles via RDRPs and their application as sonosensitizers has never been reported.

In this study, we report the fabrication of biocompatible poly(2-methacryloyloxyethyl phosphorylcholine) (PMPC)-modified TiO_2_ nanoparticles with high colloidal stability under physiological conditions using RDRPs and investigate the sonosensitizing effect of TiO_2_ particles (Figure 1). To achieve this goal, a RAFT agent containing a phosphonic acid group was newly designed; the phosphonic-acid-functionalized poly(2-methacryloyloxyethyl phosphorylcholine) (PMPC-PO_4_H_2_) was synthesized via RAFT polymerization with the RAFT agent because the phosphonic acid groups can interact more strongly with the TiO_2_ surface compared to carboxylic acids [55,56,57,58]. We clarify that the colloidal stability of PMPC-PO_4_H_2_-modified TiO_2_ nanoparticles under physiological conditions was higher than that of poly(2-methacryloyloxyethyl phosphorylcholine) (PMPC) possessing carboxylic acid (PMPC-COOH)-modified TiO_2_ nanoparticles; PMPC-COOH was prepared using a commercially available RAFT agent [4-cyano-4-(phenylcarbonothioylthio)pentanoic acid: RAFT-COOH]. The effect of the molecular weight of PMPC-PO_4_H_2_ on the colloidal stability of the 2-methacryloyloxyethyl phosphorylcholine (MPC)-modified TiO_2_ nanoparticles was also investigated in detail. Finally, the sonosensitizing activity of the PMPC-modified TiO_2_ nanoparticles was investigated.

## 2. Results and Discussion

PMPC, developed by Ishihara et al., is water-soluble and has a high biocompatibility derived from the phosphorylcholine motif of the lipid bilayer [59,60]. To modify PMPC as a biocompatible polymer on TiO_2_ nanoparticles, a new water-soluble dithiobenzoate-based RAFT agent possessing a phosphonic acid group (RAFT-PO_4_H_2_) was synthesized, where the phosphonic acid group can form a stable linkage with TiO_2_ nanoparticles. 4-Cyano-4-(phenylcarbonothioylthio)pentanoic acid *N*-succinimidyl ester was reacted with *O*-phosphoryl ethanolamine in a mixture of dimethylsulfoxide/water. The product was purified via inverse silica gel chromatography. The purification of the target RAFT agent was confirmed by ^1^H-NMR and ^13^C-NMR spectroscopy (see in Figure 1 and Appendix A). From ultraviolet-visible (UV–Vis) spectral measurements, RAFT-PO_4_H_2_ has a maximum absorbance wavelength of 305 nm with a small peak at 480 nm (see in Appendix A), indicating that RAFT-PO_4_H_2_ has a phenylcarbonothioylthio group. RAFT-COOH has a low solubility in water (even under basic conditions); however, RAFT-PO_4_H_2_ is easily dissolved in water. The high water solubility of RAFT-PO_4_H_2_ may be attributed to its phosphonic acid and amide groups.

The polymerization control capability of RAFT-PO_4_H_2_ as a control agent was investigated via RAFT polymerization of MPC by comparing the performance of RAFT-COOH. In the experiments, the target molecular weights of PMPC were regulated by changing the feed molar ratio of MPC to RAFT-PO_4_H_2_, where the feed molar ratio [MPC]/[RAFT-PO_4_H_2_] was set to 10, 20, 40, and 80. Deionized water was selected as the solvent for polymerization. The monomer conversions in all polymerizations using RAFT-PO_4_H_2_ were estimated to be approximately 99% by ^1^H-NMR (Figure 2). The number-average molecular weights (*M*_n_) of PMPC prepared using RAFT-PO_4_H_2_ (PMPC-PO_4_H_2_) were evaluated to be approximately 5600 (*M*_w_/*M*_n_:1.10), 7600 (*M*_w_/*M*_n_:1.11), 12,000 (*M*_w_/*M*_n_:1.12), and 20,000 (*M*_w_/*M*_n_:1.15) by gel permeation chromatography (GPC) when the target molecular weights were set to 3400, 6300, 12,200, and 24,000, respectively (Table 1). The difference between the experimental and theoretical *M*_n_ values was caused by the difference in the excluded volume of the polymer chains between PMPC and PEG, which was used as a standard polymer for preparing a calibration curve for GPC measurements. Methanol was selected as the solvent for RAFT polymerization of MPC with RAFT-COOH because of the low solubility of RAFT-COOH. The conversion of RAFT polymerization of MPC reached 99% at all feed molar ratios of [MPC]/[RAFT-COOH] (see in Appendix A). The *M*_n_ of PMPC prepared using RAFT-COOH (PMPC-COOH) was evaluated to be approximately 5100 (*M*_w_/*M*_n_:1.11), 6400 (*M*_w_/*M*_n_:1.11), 9800 (*M*_w_/*M*_n_:1.16), and 16,000 (*M*_w_/*M*_n_:1.21) when the target molecular weights were 3200, 6200, 12,100, and 24,000, respectively (Figure 3, Table 1). These results indicate that PMPC-PO_4_H_2_ and PMPC-COOH were successfully prepared with narrow molecular weight distributions using RAFT-PO_4_H_2_ and RAFT-COOH, respectively. Furthermore, PMPC-PO_4_H_2_ and PMPC-COOH showed absorbances at 491 nm and 488 nm, respectively, which were derived from the dithiobenzoate groups of the RAFT end groups (see in Appendix A).

For the preparation of PMPC-modified TiO_2_ nanoparticles, we used the *grafting-to* approach to prepare PMPC-modified TiO_2_ nanoparticles using PMPC-PO_4_H_2_ and PMPC-COOH. In the modification step, PMPC-PO_4_H_2_ was added to the aqueous dispersion of TiO_2_ in the presence of polyoxyethylene (20) oleyl ether (Brij98), where the pH of the aqueous media was adjusted to 4.0, to form OH groups on the TiO_2_ nanoparticles. Notably, the TiO_2_ nanoparticles were coagulated while adjusting to pH 4.0 without Bij98, whereas the particles were stably dispersed upon the addition of Brij98; this phenomenon was caused by the steric repulsion derived from the adsorbed Brij98 on TiO_2_ nanoparticles. The adsorption of Brij98 on TiO_2_ nanoparticles was supported by zeta potential measurements, that is, the zeta potential of the original TiO_2_ nanoparticles (without Brij98) at pH 4.0 was +41.5 mV, whereas the zeta potential of the TiO_2_ nanoparticles decreased slightly to +34.6 mV upon addition of Brij98, indicating that Brij98 was slightly adsorbed on the TiO_2_ nanoparticles. The Brij98-stabilized TiO_2_ nanoparticle dispersion had a monomodal particle size distribution, with a size of 38 nm (PDI: 0.256) (Figure 4). After the modification of PMPC-PO_4_H_2_ (*M*_n_: 5600) on the TiO_2_ nanoparticles, a peak derived from the submicrometer-sized coagulate TiO_2_ particles was observed with a peak derived from the non-coagulated TiO_2_ particles (average particle size: 55 nm, PDI: 0.256), indicating that the steric repulsion between TiO_2_ particles is not effective for maintaining the colloidal stability when using the short PMPC-PO_4_H_2_. When PMPC-PO_4_H_2_ of 7600 in *M*_n_ was used, a small coagulate peak was also observed [47.2 nm (PDI: 0.263) for 7600 in *M*_n_]. However, the coagulated TiO_2_ particles were not detected when using the other PMPC-PO_4_H_2_ of 12,000 and 20,000 in *M*_n_ [49.3 nm (PDI: 0.194) for 12,000 in *M*_n_, and 55.7 nm (PDI: 0.151) for 20,000 in *M*_n_]. Furthermore, the zeta potential of PMPC-modified TiO_2_ nanoparticles prepared with PMPC-PO_4_H_2_ (*M*_n_: 20,000) decreased markedly to +4.8 mV, which indicates that the modification of PMPC on TiO_2_ particles was successful. Moreover, the similar particle size distribution of PMPC-modified TiO_2_ particles prepared with PMPC-PO_4_H_2_ of 12,000, and 20,000 in *M*_n_ was maintained even after 240 min in pure water and after 100 times dilution of these particles in pure water [48.2 nm (PDI: 0.203) for 12,000 in *M*_n_, and 53.3 nm (PDI: 0.172) for 20,000 in *M*_n_ after 240 min] (see in Appendix A). Thus, colloidally stable PMPC-modified TiO_2_ nanoparticles were successfully prepared using PMPC-PO_4_H_2_ with sufficient molecular weight. A similar molecular weight effect on the TiO_2_ particle size distribution was observed when using PMPC-COOH with different molecular weights. A notable coagulation of TiO_2_ particles was detected in the particle size distribution of TiO_2_ particles incubated with PMPC-COOH of 5100 in *M*_n_. However, when PMPC-COOH with a higher molecular weight was used, particle size distributions with small peaks derived from coagulated TiO_2_ particles were observed. The zeta potential of PMPC-COOH (*M*_n_: 16,000) was approximately +17.9 mV, which is smaller than that of the original TiO_2_ particles but is higher than that of the PMPC-PO_4_H_2_-modified TiO_2_ particles. Furthermore, the average particle size of the PMPC-COOH-modified TiO_2_ particles increased [84.0 nm (PDI: 0.236) for 16,000 in *M*_n_, and 102.0 nm (PDI: 0.240) for 9800 in *M*_n_] just after 100 times dilution using pure water; a more marked increase in the particle size of PMPC-COOH-modified TiO_2_ particles just after dilution was detected using lower molecular weight PMPC-COOH, although the particle size was maintained after 240 min (see in Appendix A). The difference in the colloidal stability of PMPC-PO_4_H_2_- and PMPC-COOH-modified TiO_2_ particles during pure water dilution may be caused by the higher affinity of the PO_4_H_2_ group to the TiO_2_ particles than that of the COOH group. These results indicate that the colloidal stability of PMPC- PO_4_H_2_-modified TiO_2_ particles is higher than that of PMPC-COOH-modified TiO_2_ particles against pure water dilution.

We further investigated the effect of the PMPC-PO_4_H_2_ concentration on the particle size of obtained PMPC-modified TiO_2_ particles using PMPC-PO_4_H_2_ (*M*_n_: 7600) (see in Appendix A and Table 2). The average particle size of PMPC-modified TiO_2_ particles increased with increasing concentration from 2.5 mg/mL (41.6 nm, PDI: 0.181) to 5.0 mg/mL (49.1 nm, PDI: 0.264). However, the average particle size was almost saturated above 5.0 mg/mL, i.e., 47.2 nm (PDI: 0.263) and 49.3 nm (PDI: 0.271), and a monodispersed distribution was maintained when the PMPC- PO_4_H_2_ concentration was set to 0, 2.5, 5.0, 10.0, and 20.0 mg/mL, respectively (see in Appendix A).

For application as a sonosensitizer, PMPC-modified TiO_2_ particles with high colloidal stability under physiological conditions, including pH and ionic concentration, are required. Thus, the colloidal stability of PMPC-modified TiO_2_ particles prepared using PMPC-PO_4_H_2_ and PMPC-COOH was investigated in PBS (pH 7.4) using DLS. The particles of Brij98 stabilized TiO_2_ particles immediately coagulated in PBS (Figure 5g). PMPC-modified TiO_2_ particles prepared using PMPC-PO_4_H_2_ (*M*_n_: 5600) showed higher stability than unmodified TiO_2_ particles because the particles were not immediately coagulated in PBS. However, the particle size significantly increased with increasing incubation time, reaching 199 nm (PDI: 0.279) after 60 min of incubation. We found that the colloidal stability of the PMPC-modified TiO_2_ particles increased with the increasing molecular weight of PMPC-PO_4_H_2_. The particle sizes of PMPC-modified TiO_2_ particles after 60 min of incubation were 86.5 nm (PDI: 0.200), 77.4 nm (PDI: 0.188), and 56.7 nm (PDI: 0.159) when PMPC- PO_4_H_2_ of 7600, 12,000, and 20,000 in *M*_n_ was used, respectively. In particular, the PMPC_20,000_-modified TiO_2_ particles were maintained at less than 100 nm even after 240 min of incubation. In contrast to PMPC-PO_4_H_2_, the particle size of the PMPC-modified TiO_2_ nanoparticles increased immediately even when high-molecular-weight PMPC-COOH (9800 and 16,000) was used (Figure 5). These results strongly indicate that the phosphonic acid groups of PMPC-PO_4_H_2_ are necessary to obtain colloidally stable PMPC-modified TiO_2_ nanoparticles under physiological conditions. Previously, to prepare self-assembled monolayers (SAMs) on TiO_2_ substrates or to form modification layers of TiO_2_ photocatalysts, various molecules possessing acidic functional groups (e.g., carboxylic acid and phosphonic acid) were widely used, where these acidic groups work as interaction sites for the TiO_2_ surface [55,56,57,58]. Several groups have reported that phosphonic acids interact more strongly with TiO_2_ surfaces compared to carboxylic acids [61]. Gao et al. reported that well-ordered SAMs were formed on TiO_2_ surfaces with phosphonic acid compounds, whereas most carboxylic acid compounds were removed from the TiO_2_ surface during the washing process [62]. Thus, it appears that PMPC-COOH may be desorbed from TiO_2_ nanoparticles in the buffered aqueous solution, resulting in particle coagulation, whereas TiO_2_ nanoparticles with high colloidal stability were obtained with PMPC- PO_4_H_2_ and had a stronger interaction capability with TiO_2_.

Finally, we investigated the ^1^O_2_ generation capability of PMPC_20,000_-PO_4_H_2_-modified TiO_2_ particles under sonication in PBS using singlet oxygen sensor green (SOSG) as a probe molecule; the fluorescence intensity derived from SOSG increases upon reaction with ^1^O_2_. As shown in Figure 6, the fluorescence intensity of SOSG increased gradually with increasing sonication time and was significantly higher than that of the control sample in the absence of PMPC_20,000_-PO_4_H_2_-modified TiO_2_ particles (buffer solution). Furthermore, the fluorescence intensity derived from ^1^O_2_-reacted SOSG for PMPC_20,000_-PO_4_H_2_-modified TiO_2_ particles was higher than that for PMPC_7600_-PO_4_H_2_-modified TiO_2_ particles. These results indicate that the PMPC-PO_4_H_2_-modified TiO_2_ particles with high colloidal stability in the buffer solution exhibited ^1^O_2_ generation ability under sonication conditions in aqueous media.

## 3. Conclusions

In this study, we successfully created PMPC-modified TiO_2_ nanoparticles with high colloidal stability in PBS as novel sonosensitizers using PMPC-PO_4_H_2_. To prepare PMPC-PO_4_H_2_, a new water-soluble RAFT agent possessing a phosphonic acid group (RAFT-PO_4_H_2_) was synthesized. Using RAFT-PO_4_H_2_, PMPC-PO_4_H_2_ with a narrow molecular weight distribution was prepared. Further, the molecular weight of PMPC-PO_4_H_2_ could be regulated by changing the molar ratio [MPC]/[ RAFT-PO_4_H_2_]. The *grafting-to* approach using PMPC-PO_4_H_2_ yielded PMPC-modified TiO_2_ nanoparticles, and PMPC-PO_4_H_2_ with a higher molecular weight yielded greater colloidal stability of the PMPC-modified TiO_2_ nanoparticles. Moreover, the PMPC-PO_4_H_2_-modified TiO_2_ nanoparticles had greater colloidal stability under physiological conditions than the PMPC-COOH-modified TiO_2_ nanoparticles. Furthermore, the sonosensitizing effect of the PMPC-modified TiO_2_ nanoparticles in assisting ^1^O_2_ generation in an aqueous medium was clarified. Utilizing the RAFT polymerization, the TiO_2_ nanoparticles can be further functionalized. For example, the sonosensitizer can be further functionalized by 2nd block chain extension and/or RAFT chain end modification. We believe that the various functionalized biocompatible polymer-functionalized TiO_2_ nanoparticles will be developed for sonodynamic therapy. 

## 4. Materials and Methods

### 4.1. Materials

4-Cyano-4-(phenylcarbonothioylthio)pentanoic acid *N*-succinimidyl ester, *O*-phosphoryl ethanolamine, and MPC were purchased from Sigma–Aldrich (St. Louis, MO, USA). 2,2’-Azo*bis*[2-(2-imidazolin-2-yl)propane]dihydrochloride (VA-044) and Brij98 were purchased from Wako Pure Chemical Co., Ltd. (Osaka, Japan). NaCl, HCl, NaOH, and dimethyl sulfoxide (DMSO) were purchased from Nacalai Tesque (Kyoto, Japan). A dispersion of TiO_2_ nanoparticles (STS-100) was purchased from Ishihara Sangyo Kaisha Ltd. (Osaka, Japan). Deionized water was obtained using a Millipore Milli-Q purification system. SOSG was purchased from Thermo Fisher Scientific (Waltham, MA, USA).

### 4.2. Apparatus

UV–Vis spectral measurements were performed using a V-560 spectrophotometer (Jasco Ltd., Tokyo, Japan). Fluorescence spectral measurements were performed using an FP-8300 spectrophotometer (Jasco Ltd., Tokyo, Japan). ^1^H-NMR spectra were measured using a 400-MHz Fourier transform (FT)-NMR apparatus (JNM-ECX400, FT-NMR system, JEOL Ltd., Tokyo, Japan). The particle size distribution and zeta potential of the obtained particles were measured using a ZETASIZER NANO-ZS instrument (Malvern, UK). Ultrasonication was performed using Sonitron2000 (NEPA GENE, Chiba, Japan). The number- and weight-average molecular weights (*M*_n_ and *M*_w_, respectively) were analyzed by GPC at 40 °C using TSKgel G3000PW and TSKgel G4000PW (7.8 mm i.d. × 300 mm, Tosoh Corp.) with 20 mM phosphate buffer (pH 7.4) as the eluent, coupled with a refractive index detector (RI-2031 Plus, JASCO, Tokyo, Japan). A PEG standard (molecular weight range: 1080–107,000) was used to calibrate the molecular weight. Theoretical molecular weights were calculated using the following Equation (1). In Equation (1), *M*_n.Monomer_ and *M*_n.RAFT_ are molecular weights of monomer and RAFT agent, respectively, [Monomer] and [RAFT agent] were molar concentrations of monomer and RAFT agent, respectively.
(1)$$M_{n.\mathrm{th}}=M_{n.\mathrm{Monomer}}\times \frac{\left[\mathrm{Monomer}\right]}{\left[\mathrm{RAFT}\;\mathrm{agent}\right]}\times \mathrm{Conversion}+M_{n.\mathrm{RAFT}}$$

### 4.3. Synthesis of RAFT- PO_4_H_2_

4-Cyano-4-(phenylcarbonothioylthio)pentanoic acid *N*-succinimidyl ester (546.6 mg, 1.45 mmol) was dissolved in DMSO (30 mL). *O*-phosphoryl ethanolamine (234.6 mg, 1.66 mmol) was dissolved in a carbonate buffer (pH 9, 15 mL). These solutions were mixed to facilitate a coupling reaction between these molecules at room temperature for 18 h in the dark. The product was purified via inverse silica gel chromatography using methanol and water. Methanol and water were removed by evaporation and freeze-drying, respectively, to yield a dry product. Yield: 82%.

### 4.4. Synthesis of PMPC by RAFT Polymerization

MPC (1 mmol), RAFT-PO_4_H_2_ (10, 20, 40, 80 μmol), and VA-044 (2.5, 5.0, 10, 20 μmol) were dissolved in deionized water (3, 3, 5, 8 mL). The solution was added to a Schlenk flask. After several N_2_/degassing processes, the polymerization began with heating the solution at 40 °C for 24 h in the dark. After polymerization, the polymer (PMPC-PO_4_H_2_) was obtained by freeze-drying. To prepare PMPC-COOH, RAFT polymerization was performed under the same conditions as the prepolymer solution of methanol (3 mL) containing MPC (1 mmol), RAFT-COOH (10, 20, 40, and 80 μmol), and VA-044 (2.5, 5.0, 10 and 20 μmol).

### 4.5. PMPC-Modified TiO_2_ Nanoparticles

TiO_2_ nanoparticles were dispersed in deionized water by dissolving Brij98 (1.5 mL), and PMPC- PO_4_H_2_ or PMPC-COOH aqueous solution (1.5 mL) was mixed with the dispersion of TiO_2_ nanoparticles (final concentration:1 mg/mL TiO_2_, 0.5 mM Brij98, 10 mg/mL PMPC). After the pH was adjusted to 4.0, the mixture was incubated for 24 h at room temperature in the dark. The size distribution of the incubated particles was determined using DLS. The supernatant of the dispersion was separated by ultrafiltration (50,000 Da), and the UV–Vis spectra of the supernatant and the original PMPC aqueous solution were measured to evaluate the modification of PMPC-PO_4_H_2_ or PMPC-COOH on TiO_2_ nanoparticles.

### 4.6. Colloidal Stability of PMPC-Modified TiO_2_ Nanoparticles in PBS

The dispersion of PMPC-modified TiO_2_ nanoparticles (2.5 μL) was mixed with PBS (pH 7.4, 2.9975 mL). After several incubation periods, the particle size distributions were measured using DLS. Stabilized TiO_2_ nanoparticles were used as a reference instead of PMPC-modified TiO_2_ nanoparticles.

### 4.7. Sonosensitizing Effect of PMPC-Modified TiO_2_ Nanoparticles

The dispersion of PMPC-modified TiO_2_ nanoparticles (3 mL, TiO_2_ concentration: 45 μL/mL) was mixed with a methanol solution of SOSG (0.5 M, 6 μL). Thereafter, sonication (intensity: 0.5 W/cm^2^) was performed for 2, 4, 6, 8, and 10 min. The fluorescence intensity of SOSG was measured at excitation wavelength of 485 nm and an emission wavelength of 525 nm.

## Data Availability

The data presented in this study are available in the article.

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
