# Peer review of "Biocompatible Polymer-Grafted TiO2 Nanoparticle Sonosensitizers Prepared Using Phosphonic Acid-Functionalized RAFT Agent"

_polymers, 2023, doi:10.3390/polym15112426_

Round 1

Reviewer 1 Report

The main research content of this paper is aimed at the development and improvement of ultrasound sensitizers in ultrasound therapy. The authors prepared a novel biocompatible ultrasonic sensitizer, poly(2-methacryloyloxyethyl phosphorylcholine)(PMPC)-modified titanium dioxide nanoparticles, using phosphoric acid-functionalized reversible addition-break chain transfer agent. The authors used grafting to modify PMPC on the surface of titanium dioxide nanoparticles and demonstrated that phosphate terminal groups are essential for generating colloidal stability. During the experiment, the authors also found that this new ultrasound sensitizer can enhance the production of singlet oxygen (1O2).

  In short, this paper is somewhat innovative in the development of new ultrasound sensitizers, and its experimental data and analysis methods are also rigorous. This study has certain practical application significance for improving the efficacy and biocompatibility of ultrasound therapy.

  However, there are some errors and deficiencies in this manuscript, and my summary comments are as follows:

1.     In the preamble, the introduction of acoustic dynamic therapy is too simple (concept, advantages), and the mechanism of action of acoustic dynamic therapy in cancer treatment is not comprehensive, it is recommended to make corresponding supplements;

2.     In the preamble, the role of sonosensitizers in treatment is not explained, and it is recommended to supplement this aspect;

3.     In the preface, there are various methods to prepare highly stable titanium dioxide nanoparticles for SDT under physiological conditions. What is the necessity and importance of the highly stable titanium dioxide prepared in this study? Without specifying it, it is recommended to add;

4.     The majority of references in the last 5 years should be selected so that the advantages of this study can be obtained in the discussion section.

5.     The conclusion part is too little, lacking their own thinking and prospects;

Author Response

Thank you for your careful peer review of our manuscript. Please find the uploaded file for our answers to your comments.

Reviewer 2 Report

The paper entitled „Poly(2-methacryloyloxyethyl Phosphorylcholine)-Grafted Titanium Dioxide Nanoparticles as Biocompatible Sonosensitizers Prepared Using Phosphonic Acid-Functionalized Reversible Addition-Fragmentation Chain Transfer Agent” by Yukiya Kitayama, Aoi Katayama, Zhicheng Shao and Atsushi Harada, reports the development poly(2-methacryloyloxyethyl phosphorylcholine) (PMPC)-modified TiO2 nanoparticles as new biocompatible sonosensitizers. Their properties has been also reported. Paper has serious flaws and before reconsideration have to be addressed below listed issues:

·         The authors write: "Biocompatible polymer-modified TiO2 nanoparticles with high colloidal stability under physiological conditions using RDRPs" - no evidence of biocompatibility of the material was presented in the study.

·         The formation of TiO2 coagulums is extremely widely reported in the literature, no need for speculation (155-159)

·         Please summarize the data on the characterization of nanomaterials in tabular form. In its current (narrative) form, it is very burdensome for the recipient to analyze.

·         Data on the generation of singlet oxygen are practically undeveloped.

Therefore, I recommend to reject.

Author Response

(The authors gave the same response as above.)

Reviewer 3 Report

1. The title of the article is very cumbersome. It is desirable to simplify it considerably.

2. The text of the article compares the experimental and theoretical molecular weights of polymers. The first ones were obtained according to GPC data. How were the theoretical molecular weights of polymers calculated? This is not in the discussion of the results, nor in the experimental part.

3. In the experimental part, when describing the GPC method, the range of molecular weights of calibration samples is not given.

4. The literary references given to the article reflect the relevance of the work. Literary references should preferably be given for the last 5 years. In this article, most of the references refer to studies performed much earlier. This needs to be fixed.

5.  Notes on the design of the work:

Ø  In the text of the article, the abbreviation EPR refers to "enhanced permeability and retentivity". It is generally accepted in chemistry to apply this abbreviation to the method of analysis "electron paramagnetic resonance". This should be corrected. You can generally remove the encryption, because. the combination ““enhanced permeability and retentivity” is used once in the text.

Ø  The caption to fig. 2. It should be indicated that the scheme (a) and NMR spectra (b) are shown.

Ø  In Figure 6, everywhere and in Figure 7 a, b, two values at the origin should be removed. It is not correct.

Author Response

(The authors gave the same response as above.)

Round 2

Reviewer 2 Report

1. The answer to Q4 is not exhaustive. The reviewer is obviously aware of the mechanism of action of chemical quenchers such as SOSG, but I do not think that using only the comparative method is sufficient. Please do a quantitative analysis.

2. The general tone of the article (including even the title) suggests that the material was developed for biological, medical or biomedical use. Still no correspondence with the content. Please refrain from far-reaching comparative interpretation in the research work.

Round 3

Reviewer 2 Report

Referring to A1, the reviewer can do nothing but conclude that the described research does not present sufficient value for publication. In fact, it is a description of a relatively simple synthesis of a compound with already known biological properties (the authors tried so hard to prove it to me that I finally believed it). The sonodynamic characterization is not sufficient, as the reviewer states for the third time (though perhaps too subtly).

Author Response

In our study, we evaluated 1O2 generation by sonication using singlet oxygen sensor green (SOSG), because it was widely believed that 1O2 was most effective species in ROS for inducing cell killing effect. The reviewer commented “quantitative” analysis using SOSG as sonodynamic characterization. However, the detailed information of SOSG including chemical structure etc. was not provided from manufacture and we cannot estimate the number of fluorescent molecules generated by sonication from the change in fluorescence intensity. So, it is impossible to perform “quantitative” analysis for 1O2 generation using SOSG, which commented by the reviewer. We have already revised the manuscript from the reviewers carefully, and we believe that the revised manuscript is acceptable as an article in Polymers.